# Disruption of a DUF247 Containing Protein Alters Cell Wall Polysaccharides and Reduces Growth in *Arabidopsis*

**DOI:** 10.3390/plants12101977

**Published:** 2023-05-15

**Authors:** Pitchaporn Wannitikul, Pakorn Wattana-Amorn, Sukhita Sathitnaitham, Jenjira Sakulkoo, Anongpat Suttangkakul, Passorn Wonnapinij, George W. Bassel, Rachael Simister, Leonardo D. Gomez, Supachai Vuttipongchaikij

**Affiliations:** 1Department of Genetics, Faculty of Science, Kasetsart University, 50 Ngam Wong Wan Road, Chattuchak, Bangkok 10900, Thailand; pitchaporn.wan@ku.th (P.W.); sukhita.sa@ku.th (S.S.); jane0610@hotmail.com (J.S.); anongpat.s@ku.th (A.S.); passorn.w@ku.th (P.W.); 2Special Research Unit for Advanced Magnetic Resonance and Center of Excellence for Innovation in Chemistry, Department of Chemistry, Faculty of Science, Kasetsart University, Ngam Wong Wan Road, Chattuchak, Bangkok 10900, Thailand; fscipwa@ku.ac.th; 3Center of Advanced studies for Tropical Natural Resources, Kasetsart University, Ngam Wong Wan Road, Chattuchak, Bangkok 10900, Thailand; 4Omics Center for Agriculture, Bioresources, Food and Health, Kasetsart University (OmiKU), Bangkok 10900, Thailand; 5School of Life Sciences, The University of Warwick, Coventry CV4 7AL, UK; george.bassel@warwick.ac.uk; 6CNAP, Department of Biology, University of York, Heslington, York YO10 5DD, UK; rachael.hallam@york.ac.uk (R.S.); leonardo.gomez@york.ac.uk (L.D.G.)

**Keywords:** DUF247, mannan, pectin, plant cell wall, plant growth, xylan, xyloglucan

## Abstract

Plant cell wall biosynthesis is a complex process that requires proteins and enzymes from glycan synthesis to wall assembly. We show that disruption of *At3g50120* (*DUF247-1*), a member of the *DUF247* multigene family containing 28 genes in *Arabidopsis*, results in alterations to the structure and composition of cell wall polysaccharides and reduced growth and plant size. An ELISA using cell wall antibodies shows that the mutants also exhibit ~50% reductions in xyloglucan (XyG), glucuronoxylan (GX) and heteromannan (HM) epitopes in the NaOH fraction and ~50% increases in homogalacturonan (HG) epitopes in the CDTA fraction. Furthermore, the polymer sizes of XyGs and GXs are reduced with concomitant increases in short-chain polymers, while those of HGs and mHGs are slightly increased. Complementation using 35S:*DUF247-1* partially recovers the XyG and HG content, but not those of GX and HM, suggesting that DUF247-1 is more closely associated with XyGs and HGs. *DUF247-1* is expressed throughout *Arabidopsis*, particularly in vascular and developing tissues, and its disruption affects the expression of other gene members, indicating a regulatory control role within the gene family. Our results demonstrate that DUF247-1 is required for normal cell wall composition and structure and *Arabidopsis* growth.

## 1. Introduction

Plant cell wall biosynthesis is a key cellular process required for plant growth and development [1]. Cell wall polysaccharides are synthesized via two routes: at the cell membrane for cellulose and callose biosynthesis and in the Golgi for matrix polysaccharide biosynthesis [2]. The main mechanisms for matrix polysaccharide biosynthesis involving glycosyltransferases and sidechain substitutions are well established, and many of the genes responsible for these processes have been identified [3,4,5,6]. Other related processes and proteins are mostly unknown, including peripheral proteins or enzymes required for the matrix polysaccharide biosynthesis, the delivery to the extracellular matrix and the assembly to cellulose microfibrils and preexistent walls. Emerging evidence in these areas includes MSR accessory proteins for heteromannan (HM) biosynthesis [7,8], extensin AtEXT3-mediated wall assembly [9] and the ultrastructure of matrix cell wall polysaccharide assembly [10]. Further discovery of the proteins and enzymes involved in the route from biosynthesis to wall assembly will enhance our understanding of plant cell wall biosynthesis.

Putative cell wall-associated genes and proteins have been identified through various approaches, notably genome-wide expression and co-expression analyses using parameters related to cell wall metabolism [11,12,13,14,15] and proteomic analysis of cell wall solubilization and fractionation [16,17]. Among these, cell wall-associated proteins that contain a domain of unknown function (DUF) pose a challenge for cell wall biologists and protein biochemists. The hidden Markov model (HMM) DUFs based on the Pfam database are uncharacterized proteins that are difficult to study due to the lack of sequence homology to proteins with known functions [18]. To study the cell wall-associated DUF-containing proteins (referred to as cell wall-DUFs), a research study mined through multi-omic data sets devoted to cell wall study ranging from wall biosynthesis to wall-deposited proteins and summarized up to 105 *Arabidopsis* loci belonging to 53 cell wall-DUF families [19]. These cell wall-DUFs are good candidates for functional characterization and studying related wall processes. However, only DUF23 [20], DUF231 [21] and DUF579 [22] so far have been characterized and re-annotated among these DUF families, while a few others have been reported with unclear functions.

DUF247-containing proteins (DUF247 proteins herein) are encoded by a large multigene family of 28 coding sequences in *Arabidopsis*. While some members have been implicated in plant development, sex incompatibility and sex determination, the molecular function of the DUF247 protein family is not well understood. Overexpression of *DUF247-At3g60470* (*DLE1*) has been shown to cause severe dwarfism and bushy inflorescences in *Arabidopsis*, suggesting a potential role in a plant defense system responding to environmental stimuli and biotic stresses [23]. *SOFF* (*suppression of female function*), which is closely similar to *Arabidopsis DUF247-At4g31980*, is Y chromosome-specific for the male genome of asparagus and plays a role in suppressing pistil development in male flowers [24,25]. Additionally, two *DUF247* genes were identified through map-based cloning as key genes in the *S* and *Z* loci that regulate sex incompatibility (SI) in perennial ryegrass (*Lolium perenne* L.) [26,27,28]. Other studies suggest that DUF247 is likely involved in the plant cell wall through associations with fiber length and lignin content via genome analysis and co-expression analysis [29,30]. Moreover, the ryegrass *DUF247*-SI was shown to control the inhibition of pollen tube growth [28,31], which is formed by rapid cell expansion and cell wall biosynthesis [32,33].

Initially, we aimed to investigate the function of various cell wall-DUF families through reverse genetics. We found a substantial reduction in growth in *Arabidopsis* insertion lines of DUF247-*At3g50120* (designated as *DUF247-1* among the 28 family members). As a result, we narrowed our focus to the DUF247 family and hypothesized that it is involved in cell wall-related processes. We conducted a detailed investigation of the in vivo function of *DUF247-1* in *Arabidopsis* using two T-DNA knockout mutants and complementation through molecular study and cell wall analysis. Our results indicate that the knockout mutants had altered cell wall polysaccharide compositions, suggesting a role of DUF247-1 in matrix polysaccharide biosynthesis.

## 2. Results

### 2.1. Arabidopsis T-DNA Insertion Knockouts of DUF247-At3g50120 Exhibit Reduced Growth

A DUF247-*At3g50120* mutant line, called *duf247-1-1* hereafter, was firstly identified during our screening of T-DNA insertion mutants of cell wall-DUFs because it had slower growth and a smaller plant size than the wild type (WT), from seedlings to mature stages (Figure 1a–d). The mature mutant plants had slightly smaller rosettes compared to the WT, while the inflorescence stems were similar in height to the WT. An independent knockout line, *duf247-1-2*, shows a similar altered-growth phenotype (Figure 1a–d). Figure 2a represents the T-DNA insertions of *duf247-1-1* and *duf247-1-2* at the second exon of *DUF247-1*. The homozygosity of both mutants was verified by PCR (Figure 2b). RT-PCR at 30, 35 and 40 cycles indicated the absence of the *DUF247-1* transcript, confirming that these lines are true knockouts (Figure 2c). An analysis of seedling growth at 7 days after germination (DAG) shows that both mutant lines had ~30% shorter root lengths than the WT (*p* < 0.01), and dark-grown etiolated hypocotyls examined at 4 DAG show significantly reduced hypocotyl lengths in both lines (*p* < 0.001) (Figure 1e,f). Noting that the hypocotyl lengths were different between the mutant lines, it is unclear if this is a result of different T-DNA insertion positions. Complementation of the *duf247-1-1* mutant using 35S:*DUF247-1-FLAG* provided recovery of the growth phenotypes and confirmed that the observed phenotype was due to *DUF247-1* knockouts. Thus, complementation and the two independent T-DNA insertion lines demonstrated that the disruption of the cell wall-associated DUF247-*At3g50120* caused reduced growth in *Arabidopsis.*

### 2.2. DUF247-1 Knockouts Cause Altered Cell Wall Composition

We examined the *duf247-1* mutants for potential cell wall alterations by analyzing matrix polysaccharides in seven-day-old seedlings. A monosaccharide composition analysis of the hot water–CDTA fraction shows a two-fold increase in GalA, slight increases in Rha and GluA and decreases in Gal and Glu in the mutants, whereas that of the TFA (trifluoroacetic acid) fraction did not show a clear change (Appendix A). Complementation using 35S:*DUF247-1* showed altered monosaccharide profiles both in the hot water–CDTA and TFA fractions, suggesting a role of DUF247-1 in modifying cell wall matrix polysaccharides. Fourier-transform infrared (FTIR) analysis of AIR samples shows that the absorbance of wavenumbers associated with various cell wall components in the mutants, including cellulose, hemicellulose and pectin, are reduced across the spectrum and that of the complemented line shows a recovery to the level observed in the WT (Figure 3). However, as the mutants grow slower than the WT, it is important to note that the observed changes in the overall FTIR profile of the mutants could be attributed to both developmental differences of the seedlings and the effect of the mutation on the cell wall composition. These results indicate that there are general changes across cell wall components as a result of *DUF247-1* gene disruption in *duf247-1* mutants.

We further analyzed the matrix polysaccharide composition by ELISA using glycan-directed mAbs (LM15 for xyloglucan (XyG), LM21 for heteromannan (HM), LM28 for glucuronoxylan (GX), LM19 for homogalacturonan (HG) and LM20 for methyl-esterified HG (mHG)). NaOH extracts were used for XyG, HM and GX analysis, while CDTA extracts were used for HG analysis. We found that both *duf247-1* mutants had approximately 50% reductions in XyG, HM and GX epitopes and ~50% increases in HG and methyl-esterified HG epitopes in comparison to those of the WT (*p* < 0.001) (Figure 4). Interestingly, the *duf247-1-1* mutant line complemented with the *DUF247-1* gene under the 35S promoter, showed recovery of XyG, HG and methyl-esterified HG epitopes, but not for HM and GX epitopes. This result suggests that the native promoter of *DUF247-1* may be required for full complementation of the polysaccharide phenotype. The recovery of XyG and HG epitopes in the complemented lines suggests that DUF247-1 is more associated with XyGs and HGs rather than GXs and HMs. These results indicate that *DUF247-1* knockouts cause simultaneous alterations in different cell wall matrix polysaccharides.

### 2.3. The Disruption of DUF247-1 Alters the Structure of Matrix Polysaccharides

Following the reductions in LM15, LM21 and LM28 epitopes and increases in LM19 and LM20 epitopes, we analyzed the structures of the XyGs, HMs, GXs and HGs in the *duf247-1* mutants and complemented lines using polysaccharide analysis with carbohydrate gel electrophoresis (PACE), based on specific glycan-hydrolases (endo-glucanase II, endo-1,4-β-xylanase, endo-1,4-β-mannanase and endo-PGase for XyGs, GXs, HMs and HGs, respectively). However, based on the PACE profiles, we did not observe alterations in the structures of these polysaccharides (Appendix A). We further analyzed the mass distribution profiles of these polysaccharides in *duf247-1-1* mutants using gel permeation chromatography followed by an enzyme-linked immunosorbent assay (GPC-ELISA). The mass profiles confirm the reduction in XyG and GX content in the mutants as detection signals from the high to medium M_r_ are generally reduced (*p* < 0.05), while those high M_r_ fractions of HGs and mHGs support the increases in the molecular sizes and perhaps quantities of both polysaccharides (Figure 5). Only slight changes in the overall profile were observed for HMs. Interestingly, the signals for the low M_r_, from fraction 35 for XyGs and fraction 22 for GXs onwards, show significant increases (*p* < 0.05), indicating the increase in small M_r_ XyGs and GXs in the mutant. This was also observed in the HM profile from fraction 32 onwards. Our results demonstrate that, besides the general alteration of XyG, HM, GX, HG and mHG content, the *DUF247-1* knockout alters the mass distribution profile of these polysaccharides by having increases in low M_r_ XyGs, GXs and HMs and increases in high M_r_ HGs and mHGs in the mutant cell wall.

### 2.4. Arabidopsis DUF247 Multigene Family Contains 28 Gene Coding Sequences

*AtDUF247-1* encodes a 531-amino acid protein. A sequence analysis based on the Pfam database presents a predominant DUF247 domain that makes up approximately 75% of the sequence and a transmembrane domain at the C-terminus (Figure 6a). This transmembrane domain represents the type I membrane-bound protein model, suggesting that the localization and function of DUF247-1 are associated with a membrane system. *AtDUF247-1* is a member of the *DUF247* multigene family consisting of 28 paralogous coding sequences in *Arabidopsis*. The *DUF247* family is widespread in land plants as a large multigene family, for example, including 73 genes in japonica rice and 22 genes in *Marchantia polymorpha*. From the databases searched, only *Physcomitrella patens* has a single *DUF247* gene in its genome. Interestingly, apart from land plants, the DUF247 domain is found only in three species of arbuscular mycorrhizal fungi, which are known to perform endosymbiosis within plant cells. A phylogenetic tree representing the evolutionary relationship among the *AtDUF247* family using amino acid sequences shows that the members are split into three groups: 17 genes for clade I, nine genes for clade II and individual *At3g02645* and *At3g03890* genes (Figure 6b). Clade I members are distributed throughout the genome, while the nine members of clade II, including *DUF247-1* (*At3g50120*), are tightly clustered on chromosome 3, potentially as a result of gene duplications. The single *PpDUF247* gene is presented as an outgroup suggesting its first emergence in the land plants before the evolutionary expansion as a gene family. This indicates that *DUF247-1* belongs to the highly similar and tightly clustered genes in clade II of the *DUF247* multigene family, encoding both functionally diverse and redundant proteins.

### 2.5. Temporal and Spatial Expression of DUF247-1 Presents a Potential Functional Redundancy within the Family Members

We investigated the expression profiles of *DUF247-1* in various Arabidopsis tissues using an Arabidopsis RNA-seq database [34] and found that it is generally expressed in the main vegetative tissues of Arabidopsis, with lower expression levels in the pollen, seed, embryo and endosperm (Figure 7). Specifically, *DUF247-1* expression levels were found to be high in the leaf, seedling, shoot and flower, while appearing to be lower in stem and meristem tissues. To gain more insight into tissue specific expression patterns, we performed a promoter–GUS fusion using the 2000 bp region upstream of *DUF247-1* to drive GUS expression. GUS activity under the *DUF247-1* promoter is generally observed throughout the plant. The expression is high in young tissues, including expanding cotyledons, shoot tips, young leaves, hypocotyls, roots, developing flowers and developing siliques, but not in the root tips (Figure 8). The expression is specific to the meristematic tissues in the root, hypocotyls and leaves, with no observed expression in the stem except in areas close to the floral tip, similar to the RNA-seq profile. *DUF247-1* expression is likely related to the developmental stage of the plant, except for secondary growth. This expression pattern is consistent with the reduced growth phenotype observed in the *duf247-1* mutants.

Despite the fact that the *DUF247* multigene family includes 28 genes with potential functional redundancy, *duf247-1* mutants display altered growth and cell wall phenotypes. We analyzed the transcript abundance of the *DUF247* clade II members by qRT-PCR using seven-day-old seedlings from the WT and *duf247-1-1* mutant (Figure 9). In the WT, the *DUF247-1* expression level is relatively low compared to other members of the gene family, with *DUF247-4* and *DUF247-5* transcripts being predominant. We note that the expression of *DUF247-1* observed in the *duf247-1-1* mutant was obtained from T-DNA truncated transcripts. In the *duf247-1-1* mutant, the transcript levels of other members are altered. *DUF247-3* and *DUF247-9* have at least four-fold increases, while *DUF247-5* has a 60% increase. Interestingly, the transcript levels of *DUF247-4* are reduced by four-fold in the mutant. This result indicates that although *DUF247-1* is not the highest expressed gene of the clade II members, the disruption of *DUF247-1* causes alterations in the transcription regulation of other gene members. However, it is also possible that the differences in transcript levels observed here could be contributed by differences in developmental stages between the WT and mutant seedlings. Furthermore, this also suggests that the function of DUF247-1 may be unique compared to other family members, as the upregulation of at least three family members could not compensate for the phenotypic alterations.

### 2.6. Predicted Structure and Function of DUF247-1

Based on the predicted structure of DUF247-1 from the AlphaFold Protein Structure Database [35,36], the protein appears to have a bundle of α-helices with a small portion of β-strands and extended N- and C-terminal helices (https://alphafold.ebi.ac.uk/entry/A0A384KQI3 accessed on 5 March 2023) (Appendix A). The latter could serve as a transmembrane domain anchoring the protein to the Golgi membrane. This is consistent with the predicted C-terminus transmembrane domain in the Pfam database. To explore the potential function of DUF247-1, a protein structure similarity search was performed using the MADOKA web server [37]. This analysis was performed on the predicted DUF247-1 structure without the extended helices at the N- and C-termini. The result showed low similarity to several protein structures in the database, including a yeast class I α-1,2-mannosidase (PDB ID: 1DL2), the only structure related to processing glycosides. This enzyme is involved in *N*-glycan processing during glycoprotein biosynthesis and its structure contains an (αα)_7_-barrel fold, where two sets of seven parallel α-helices form inner and outer barrels [38]. Despite the low similarity in structure, some of the residues involved in glycan binding are highly conserved between the yeast α-1,2-mannosidase and DUF247-1 (Figure 10). These residues were shown to interact with mannoses of α-1,2, α-1,3 and α-1,6 branches of the *N*-glycan of Man9GlcNAc2. However, these conserved residues identified in the DUF247-1 model are distributed throughout the structure and do not form a binding pocket as found in α-1,2-mannosidase. This may be attributed to the low quality of the model as these residues are located within regions characterized by low to very low per-residue confidence scores. Based on sequence alignment, this finding suggests a potential for DUF247-1 to bind with glycans and be involved with the cell wall polysaccharides.

## 3. Discussion

Our results show that disruptions to *DUF247-1* cause defects in *Arabidopsis* growth, with reduced plant size and changes in cell wall composition. These changes include ~50% reductions in XyG, HM and GX epitopes and structural changes in the mass distribution profiles of XyGs and GXs towards the low M_r_ and HGs and mHGs towards the high M_r_. These results indicate an impairment in the biosynthesis of matrix polysaccharides. A promoter–GUS fusion shows *DUF247-1* expression throughout the plant, particularly in developing and meristematic tissues, but not in the stem. qRT-PCR in WT and *duf247-1-1* seedlings shows that, although *DUF247-1* is not the predominantly expressed gene in the *DUF247* gene family, the expression profiles of the clade II are altered in the mutant, suggesting a regulatory response among the family members. Molecular characterization of the *DUF247* family so far has been reported only for the members of the clade I family [23,24,25,26,27,28], and none of these has yet been shown to be associated with the plant cell wall. Here, we show that *DUF247-At3g50120*, a member of the DUF247 family clade II, encodes a protein that plays a role related to cell wall matrix polysaccharide biosynthesis and is required for normal cell wall composition and structure, as well as normal *Arabidopsis* growth.

A high degree of functional redundancy is expected in the *DUF247* multigene gene family, although some members have distinct functions. In our case, the nine members of clade II are highly similar and are tightly clustered genes on chromosome 3 (Figure 6), which supports the case for redundancy. Yet, the phenotypic alterations are apparent in the *duf247-1* single mutants. Based on the GUS assay, the widespread expression of *DUF247-1*, especially in young and developing tissues, could account for the reduced growth and plant size seen in the mutants. Moreover, qRT-PCR of the gene cluster in WT and mutant seedlings suggests either or both of these cases: the function of *DUF247-1* is distinct from that of other clade II members based on its spatial and temporal gene expression, or the mutant phenotype is due to altered expression profiles of other clade II members. More detailed characterization of this gene family through multiple gene knockouts would help to resolve this issue. Nonetheless, the fact that *DUF247s* are present as a large gene family of 28 genes in *Arabidopsis* (and even larger numbers in other land plant species) supports that these proteins ought to play an important role in land plants [39].

Based on the Pfam database, the DUF247 family is present almost uniquely in land plants, and only three sequences are found in three species of arbuscular mycorrhiza fungi [40]. Since these fungi are known to form mutualistic symbioses with diverse land plants [41], the three fungal sequences could be derived from the land plants through horizontal gene transfer, as is reported to be the case for many other genes [42]. DUF247 may play a role in plant–microbe interactions or symbioses, which have been shown to require a number of cell wall-related proteins to mediate in these processes [43,44].

The cell wall alterations shown in this work are mostly consistent between the two independent *duf247-1* mutants and are supported by complementation experiments. A broad reduction in FTIR profiles between 800 and 1800 wavenumbers corresponding to cellulose, hemicellulose and pectin indicates a general decrease in cell wall polysaccharides, and this reflects that DUF247-1 could play a role in plant cell wall biosynthesis. Furthermore, the simultaneous reductions in the three hemicellulose polymers in the NaOH fraction and the increased HGs in the CDTA fraction in the mutants indicate that disruption of DUF247-1 causes alteration in matrix polysaccharide biosynthesis. The reduced M_r_ of XyGs and GXs and increased M_r_ of HGs and mHGs shown by GPC-ELISA indicate that alterations in the biosynthesis may occur in complex forms among different polymers. Although the increases in HG epitopes shown by ELISA and GPC-ELISA are consistent with the increases in GalA in the hot water–CDTA fraction from both mutants, they are in contrast with the FTIR data that showed a general decrease in pectins. Furthermore, the reductions in XyGs, GXs and HMs are not reflected in the monosaccharide composition in the TFA fraction. To the best of our knowledge, we are unable to interpret the monosaccharide composition and FTIR data further. The structures of matrix polysaccharides are complex, and ELISA results may not fully address the changes in these matrix polysaccharides due to limitations of the antibodies based on specific glycan structures. Further study using diverse cell wall antibodies from LM, JIM and CCRC series [45] would help resolve changes in cell wall polysaccharides in more detail. For example, JIM7 and JIM8 could help determine partial mHGs in the structure, unlike the clear discrimination between HGs and mHGs by LM19 and LM 20, respectively. Additionally, LM5 for galactan and LM6 for arabinan could verify changes in the macrostructure of pectin. Changes in the cell wall of *duf247-1* mutants appear complex as complementation could recover XyG and HG content but not the other two hemicelluloses. This evidence supports that the defects in the *duf247-1* mutants are related to the biosynthesis of matrix polysaccharides, predominantly XyGs, GXs and HGs, though the mechanism is unclear. It is important to note that concomitant alterations of different polysaccharides in a mutant are common. For example, in a mutant lacking *irx8*, a putative galacturonosyltransferase for pectin biosynthesis, a deficiency in GXs and HGs is observed [46]. Therefore, the simultaneous reductions in the three hemicelluloses and increases in HGs and mHGs could also result from a defect in one of these polymers.

Given that complementation restored normal growth and XyG and HG content, it is potentially the case that the reduction in XyGs is the cause of the growth phenotype. Reduced plant size and growth were also observed in the *xxt1 xxt2* mutant that has no detectable XyG [47]. However, it is unclear whether the growth phenotype directly resulted from the altered XyG content or whether it was a collective contribution from changes in hemicelluloses and HGs.

In recent years, a number of cell wall-associated DUFs have been reported through different approaches. DUF23, DUF246 and DUF266 were initially identified as cell wall glycosyltransferases via bioinformatic analyses [48,49] and later confirmed to be rhamnogalacturonan I (RG-I) β-1,4-galactan synthase [20], RG-I arabinogalactan biosynthesis enzymes [50] and β-glucuronosyltransferase for type-II arabinogalactan biosynthesis [51,52], respectively. Overexpression of a member of DUF231 containing proteins in *Populus* was shown to increase cellulose content, reduce recalcitrance and enhance plant growth [53]. Members of DUF231 were identified as XyG-specific *O*-acetyltransferases [21] and mannan-specific *O*-acetyltransferases [54]. DUF579 was shown to catalyze 4-0-methylation of glucuronic sidechains of GXs [22]. REL2 (a DUF630 and DUF632 containing protein), which is localized in the cell membrane, was shown to play a role in the leaf morphology of rice [55]. A DUF1218, a land plant-specific gene, was shown to be targeted to the cell membrane, and a mutant study showed that this family is involved in lignin biosynthesis [56]. Finally, AtDUF642s are associated with the cell wall through their binding capacity to cellulose and hemicellulose [57] and to PME3 (pectin methyl esterase) [58]. Here, we also used a predicted 3D structure of DUF247-1 based on the AlphaFold protein structure to investigate its molecular function. We have so far identified conserved amino acid residues in DUF247-1 that suggest its function as a mannosidase enzyme, which interacts with mannose branches of *N*-glycans of glycoproteins. This suggests that DUF247-1 functions in association with glycans in addition to its role in altering the cell wall matrix polysaccharides. Our findings add the DUF247 family to the list of cell wall-associated DUFs verified in planta.

Evidence regarding the function of DUF247-1 gathered so far is that *DUF247*-1 disruptions simultaneously cause altered amounts and structures of matrix polysaccharides. Furthermore, widespread expression of *DUF247-1* and the regulation control within the gene family may suggest the important role of the DUF247 family. Based on this evidence, we postulate that DUF247-1 is involved in the process of matrix polysaccharide biosynthesis and is required for a normal cell wall composition and *Arabidopsis* growth.

## 4. Materials and Methods

### 4.1. Plant Materials and Growth Condition

*Arabidopsis* T-DNA insertion mutant lines, *duf247-1-1* (SAIL_1252_B04) and *duf247-1-2* (SAIL_382_A09), were obtained from NASC. The *Arabidopsis* seeds were germinated on 1% (w v^−1^) agar plates containing ½ strength Murashige and Skoog (½ MS) medium and 1% (w v^−1^) sucrose for seven days and then grown in compost under the same conditions: 16 h light (125 μmol photons m^−2^ s^−1^) at 25 °C. For hypocotyl experiments, seedlings were grown on agar plates for four days in the dark. Plant materials for cell wall extraction were obtained from seedlings grown in liquid ½ MS media supplemented with 1% (w v^−1^) sucrose with constant shaking at 80 rpm for seven days.

### 4.2. Genotypic Analysis of T-DNA Insertion Mutants

gDNA was isolated by the CTAB method using 2–3 young leaves. Homozygous insertion lines for *duf247-1-1* and *duf247-1-2* were identified using 1252-F/1252-R primers and 382-F/382-R primers, respectively, for gene-specific amplification, and 1252-R/SAIL-LB and 382-R/SAIL-LB, respectively, for flanking left-border amplification (see primers in Appendix A). PCR was performed using 50 ng of gDNA in a 20 µL reaction volume containing 4 mM dNTPs, 30 mM MgCl_2_, 0.5 µM for each primer and 1 unit of *Taq* polymerase (Vivantis, Shah Alam, Malaysia) using 35 cycles of 94 °C for 1 min, 55 °C for 1 min and 72 °C for 2 min, with a final extension for 5 min. Individual plants identified as negative for the gene-specific amplification and positive for the left-border amplification were considered homozygous line candidates. Seed progenies derived from the candidates were re-tested with PCR before being designated as homozygous lines.

Total RNA was isolated from 7-day-old seedlings grown on an agar plate using TRIzol (Invitrogen, Carlsbad, CA, USA) and treated with DNase I (New England BioLabs^®^Inc., Ipswich, MA, USA) at 37 °C for 1.5 h, followed by phenol: chloroform extraction and ethanol precipitation. One microgram of total RNA was used as a template for first strand cDNA synthesis using MMuLV reverse transcriptase (Biotechrabbit, Berlin, Germany) and oligo(dT) primers. The reaction was diluted 2.5 fold and 1 µL was used for an RT-PCR reaction as follows: 4 mM dNTPs, 30 mM MgCl_2_, 0.5 µM of 382-F and 1252-R primers and 1 unit of *Taq* polymerase (Vivantis) in 20 µL reaction volume through 30, 35 or 40 PCR cycles of 94 °C for 1 min, 55 °C for 1 min and 72 °C for 2 min, with a final extension for 5 min.

### 4.3. Complementation Study

A *DUF247-1-FLAG* overexpression cassette, under the control of a 35S CaMV promoter, was constructed with the pCXSN vector [59]. The coding sequence of *DUF247-1* was amplified from cDNA using CDS-F and CDS-R primers before being re-amplified using licCDS-F and licCDS-FLAG-R primers (Appendix A). The PCR product was gel-purified and inserted into the BamHI sites by ligation-independent cloning using T4 DNA polymerase (New England BioLabs^®^Inc.). The construct was confirmed by Sanger sequencing. The *duf247-1-1* mutant was transformed by floral dipping and Hygromycin B selection (35 µg mL^−1^). Complemented T_1_ lines were confirmed using 35S and NOS primers (Appendix A). Homozygous complemented lines were identified at the T_3_ generation by genetic segregation of the seedling phenotype and 100% amplification of the expression construct from at least 15 randomly selected seedlings. Three independent complemented lines were used for the complementation study.

### 4.4. Cell Wall Preparations

Alcohol insoluble residues (AIRs) were prepared from seven-day-old seedlings by grinding in liquid nitrogen, washing three times in 80% ethanol, absolute ethanol, acetone and methanol, in that order, and then drying in an incubator at 55 °C. AIRs were extracted with 50 mM CDTA pH 7.0 at room temperature, with shaking for 18 h before filtering using nylon mesh. The residue was subsequently extracted with 4 M NaOH containing 1% (w w^−1^) NaBH_4_ at room temperature with shaking for 18 h. The soluble fraction was collected by filtering using a nylon mesh before neutralizing using glacial acetic acid. Both soluble fractions were dialyzed against distilled water at room temperature and then freeze-dried. The remaining NaOH insoluble residue was dried at 55 °C. FTIR spectroscopy was performed using AIR samples. Spectra were collected using a PerkinElmer Spectrum 100 FT-IR between 4000 and 850 cm^−1^ with resolution 4 cm^−1^ at 64 scans. Wavenumbers were assigned according to previous reports [60,61,62,63].

### 4.5. Monosaccharide Compositions of Hot Water–CDTA and TFA Fractions

The hot water–CDTA fraction was obtained by a sequential extraction of AIR samples using deionized water at 120 °C for 1 h and then 50 mM CDTA pH 6.5 at room temperature overnight. The resulting pellet was regarded as the TFA fraction. The hot water–CDTA fraction was precipitated and washed twice using 80% ethanol and dried using a vacuum evaporator. The fractions were hydrolyzed with 2 M trifluoroacetic acid at 100 °C for 4 h before separation with high-performance anion-exchange chromatography on a Dionex Carbopac PA-10 column with pulsed amperometric detection. Separated monosaccharides were quantified by external calibration using an equimolar mixture of nine monosaccharide standards (arabinose, fucose, rhamnose, xylose, glucose, galactose, mannose, glucuronic acid and galacturonic acid).

### 4.6. Polysaccharide Quantification Using Glycan-Directed Monoclonal Antibodies via ELISA

The ELISA was performed based on [45]. Glycan-directed mAbs used in this work included LM15 (XyGs) [64], LM19 (un-esterified HGs) [65], LM20 (methyl-esterified HGs) [65], LM21 (HMs) [66] and LM28 (GXs) [67]. Fifty microliters of CDTA or NaOH extracts (10 ng) were loaded onto flat-bottom 96-well plates (Costar 3599, Corning, New York, NY, USA) and dried by incubating at 37 °C overnight. The coated wells were blocked with 200 µL of 3% (w v^−1^) bovine serum albumin (BSA) in phosphate-buffered saline (PBS) at 37 °C for 1 h, probed with 25 µL of 1:50 dilution of mAb in 1% (w v^−1^) BSA in PBS at 37 °C for 1 h and washed three times with 300 µL of PBS. Subsequently, the plate was incubated with 50 µL of anti-rat IgG–HRP conjugated at 1:5000 dilution in 1% (w v^−1^) BSA in PBS at 37 °C for 1 h, washed six times with 300 µL of PBS and detected using a 3,3′,5,5′-tetramethylbenzidine (TMB) substrate solution (Vector Laboratories, Burlingham, CA, USA) for 20 min, and 50 µL of 0.5 N sulfuric acid was used to terminate the reaction. Absorbance reads at 450 and 655 nm were used for quantification against 4PL curves generated by the polysaccharides’ standards as shown in [68]. The standards, including tamarind xyloglucan (Dainippon Pharmaceutical Co., Osaka Japan), 4-O-methyl-D-glucurono-D-xylan (M5144, Sigma, St. Louis, MO, USA), guar galactomannan (P-GGM21, Megazyme, Bray, UK, Australia), polygalacturonic acid from citrus pectin (P-PGACT, Megazyme) and apple (76282, Sigma), were analyzed in a range of concentrations (3.2 ng mL^−1^ to 2 µg mL^−1^) in triplicates in the same sample plate.

### 4.7. PACE

A polysaccharide analysis using carbohydrate gel electrophoresis (PACE) was carried out following [69]. Briefly, NaOH extracts (5 mg mL^−1^) were treated with 1 unit of endo-1,4-β-xylanase from *Neocallimastix patriciarum* (E-XYLNP, Megazyme), endo-glucanase II from *Aspergillus niger* (E-CELAN, Megazyme) or endo-1,4-β-mannanase from *A. niger* (E-BMANN, Megazyme) at 40 °C for 30 min. The reactions were stopped by boiling for 10 min before drying using vacuum centrifugation at room temperature. Glucuronoxylan, tamarind XyGs and guar galactomannan, all at 2.5 mg mL^−1^ concentration, were used as standards. Oligosaccharide markers including xylobiose, xylotriose, xylotetraose, xylopentaose and xylohexaose were obtained from Megazyme. For derivatization, each sample was mixed with 5 µL of 0.1 M ANTS (2-aminonaphthalene trisulfone) (Biotium, Fremont, CA, USA) and 5 µL of 1 M NaCNBH_3_, incubated at 37 °C overnight, dried using vacuum centrifugation at 40 °C for 2 h and then resuspended in 100 µL of 6 M urea. Samples (5 µL each) were resolved in 25% (w v^−1^) polyacrylamide gel with 0.1 M Tris–borate buffer pH 8.2 at 700 V for 4 hr. The gel was visualized using a UV transilluminator. Bands representing oligoglycans were assigned based on previous reports [70,71,72] and oligosaccharide standards.

### 4.8. GPC-ELISA

GPC-ELISA was performed using the method previously described [68]. NaOH extracts were dissolved in ultrapure water at a concentration of 10 mg mL^−1^, incubation at 60 °C for 1 h before centrifugation at 15,500 g for 10 min and filtered through a 0.45 µm^2^ polyethersulfone membrane. Five hundred microliters of filtered samples (2 mg mL^−1^) were loaded onto a Superose 6 Increase 10/300 GL column (GE Healthcare, Chicago, IL, USA) on an HPLC system (Agilent/Varian Prostar 210, Palo Alto, CA, USA). Degassed ultrapure water was used as a mobile phase at a flow rate of 0.5 mL min^−1^ at 25 °C. Fractions were collected at the injection time point at 0.5 mL per fraction using a OMNICOLL fraction collector (LAMBDA instruments, Baar, Switzerland) onto 96-deep-well plates. Fifty microliters of each fraction were coated onto flat-bottom 96-well plates (Costar 3599, Corning) and incubated at 37 °C overnight. The plates were then used for ELISA using glycan-directed antibodies as described above.

### 4.9. Sequence Analysis and Phylogeny

Amino acid sequences of the *Arabidopsis* DUF247 family were selected based on the Pfam database [40] and *Arabidopsis* genome annotation [73] and retrieved from Phytozome [74]. The multiple sequence alignment was manually edited using AliView [75]. The optimum amino acid evolutionary model was identified by the maximum likelihood model selection in MEGAX [76]. The phylogenetic tree of the DUF247 family was built based on the maximum likelihood using raxmLGUI [77] with the JTT + G + I + F model and 1000 bootstrap replicates. The Protein 3D alignment search was performed using the MADOKA web server [37], and protein structure and sequence alignments were performed using PyMOL (The PyMOL Molecular Graphics System, Version 2.0 SchrÖdinger, LLC, New York, NY, USA). The sequence alignment was rendered with the ESPript web server [78].

### 4.10. Promoter–GUS Fusion and qRT-PCR

The promoter–GUS fusion cassette was constructed using 2.0 kb of the 5′ promoter region of *DUF247-1* with pCXGUS-P [58]. The promoter was PCR-amplified from *Arabidopsis* gDNA using D247_2P-F and D247_2P-R2 primers (Appendix A) and inserted into pCXGUS-P (linearized with XcmI) by a TA ligation method, and the construct was validated by DNA sequencing. The transformation was performed by floral dipping and hygromycin selection (35 µg mL^−1^). GUS expression was monitored using T_2_ plants infiltrated with GUS staining solution (0.5 mM K_3_Fe(CN)_6_, 0.5 mM K_4_Fe(CN)_6_, 60 mM Na_2_HPO_4_, 30 mM NaH_2_PO_4_, 0.1% (*w*/*v*) Triton-X100 and 0.05% (*w*/*v*) X-Gluc) and incubated at 37 °C overnight. DNase I-treated total RNA isolated from seven-day-old seedlings was used for qRT-PCR analysis. Specific primers for qRT-PCR were obtained from AtRTPrimer (http://atrtprimer.kaist.ac.kr/blan/genoPP.pl accessed on 02 April 2021) [79] or designed using Primer3 [80] (Appendix A). Primers were tested and selected based on their specificity by performing qRT-PCR reactions and agarose gel electrophoresis. qRT-PCR reactions were performed using the Luna Universal qPCR Master Mix (New England BioLabs^®^Inc.) following the manufacturer’s instructions.

## Figures and Tables

**Figure 1 plants-12-01977-f001:**
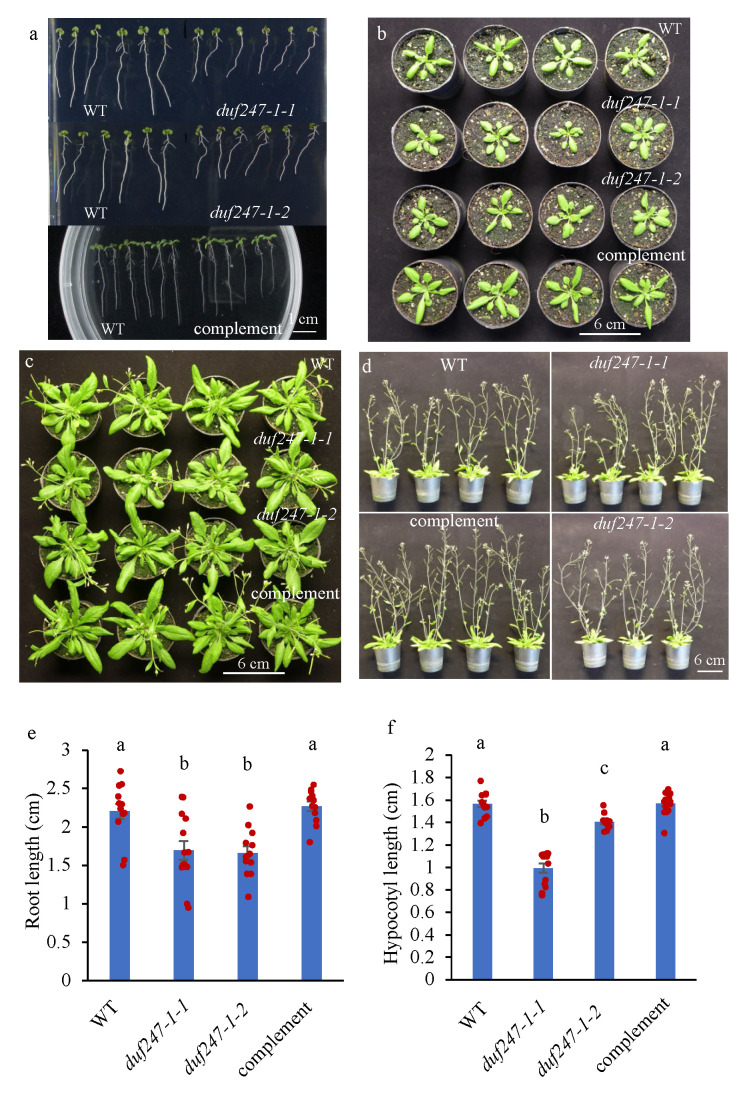
Phenotypic characterization of *duf247-1* knockout mutants. (**a**–**d**) WT, *duf247-1-1*, *duf247-1-2* and complemented lines. Plants were grown for 7 days, 3, 4 and 5 weeks, respectively. (**e**,**f**) Bars and scatter plots show root lengths at 7 DAG (n = 18) and hypocotyl lengths at 4 DAG (n = 20) from two independent *duf247-1* mutants and complemented *duf247-1-1* lines. Data are presented as mean ± SE. One-way ANOVA followed by a Tukey’s test was performed using R statistical software (version 4.1.0). Means with the same letter are not significantly different based on the Tukey’s HSD test (*p* < 0.05).

**Figure 2 plants-12-01977-f002:**
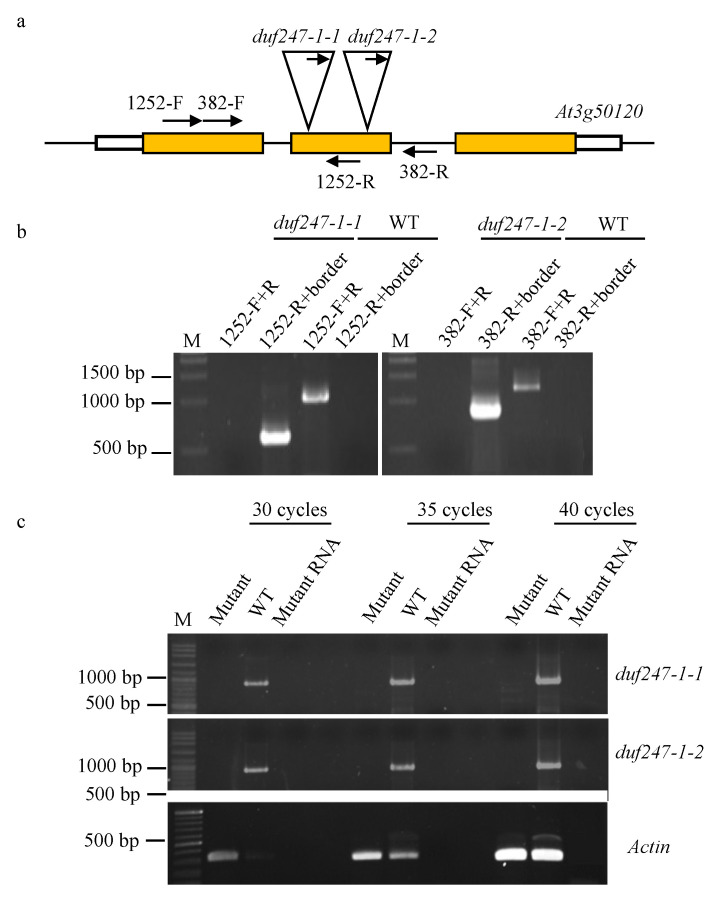
Molecular analysis of *duf247-1* T-DNA insertion mutants. (**a**) T-DNA insertion locations are indicated within the *DUF247-1* gene structure. Arrows indicate PCR primers for genotyping. (**b**) Identification of homozygous lines of *duf247-1-1* (SAIL_1252_B04) and *duf247-1-2* (SAIL_382_A09) by flanking primers and border primers. (**c**) Analysis of the *DUF247-1* transcript of *duf247-1* mutants by RT-PCR using 382-F and 1252-R primers at 30, 35 and 40 cycles. Primer locations are indicated in the gene structure. Mutant RNA loading was used as a control for no gDNA contamination. M indicates DNA ladder.

**Figure 3 plants-12-01977-f003:**
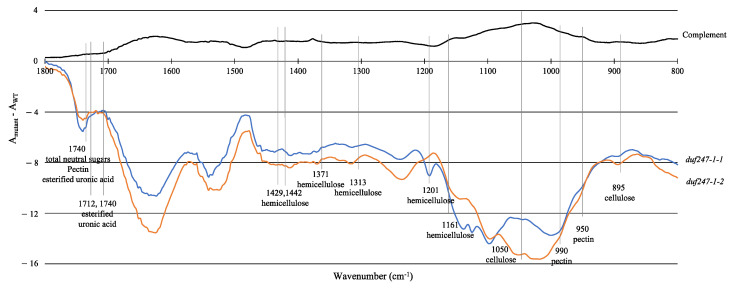
Fourier-transform infrared (FTIR) spectroscopy of AIR samples from 7-day-old seedlings of WT, *duf247-1* mutants and complemented lines. Data present average absorbance values of three reads from each sample subtracted from those of the WT.

**Figure 4 plants-12-01977-f004:**
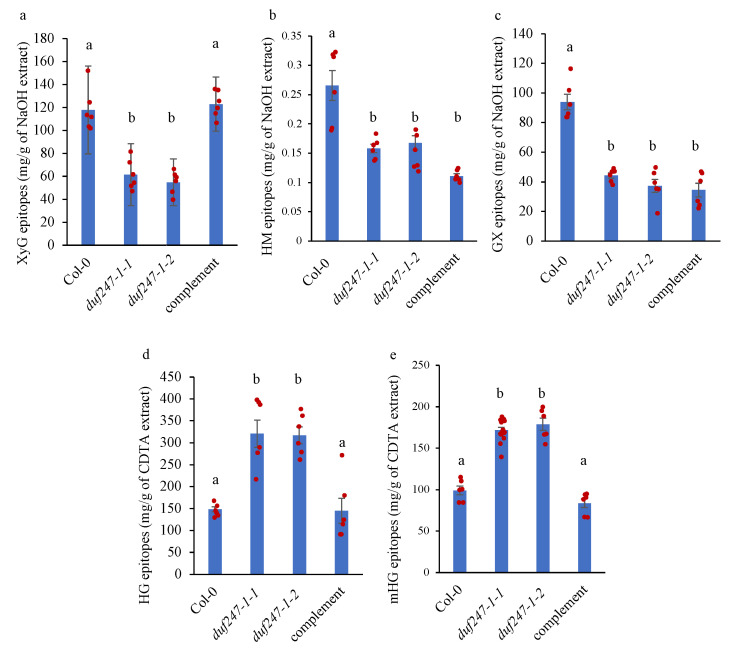
Analysis of cell wall polysaccharide epitopes by ELISA. (**a**–**c**) XyG, HM and GX epitopes in NaOH extracts were analyzed using LM15, LM21 and LM28 mAbs, respectively. (**d**,**e**) HG and methyl-esterified HG epitopes in CDTA extracts were analyzed using LM19 and LM20 mAbs, respectively. Epitope quantities were obtained from 4PL curves generated by polysaccharide standards. Data are presented as mean ± SE (n = 3) with scatter plots. One-way ANOVA followed by a Tukey’s test was performed using R statistical software (version 4.1.0). Means with the same letter are not significantly different based on the Tukey’s HSD test (*p* < 0.05).

**Figure 5 plants-12-01977-f005:**
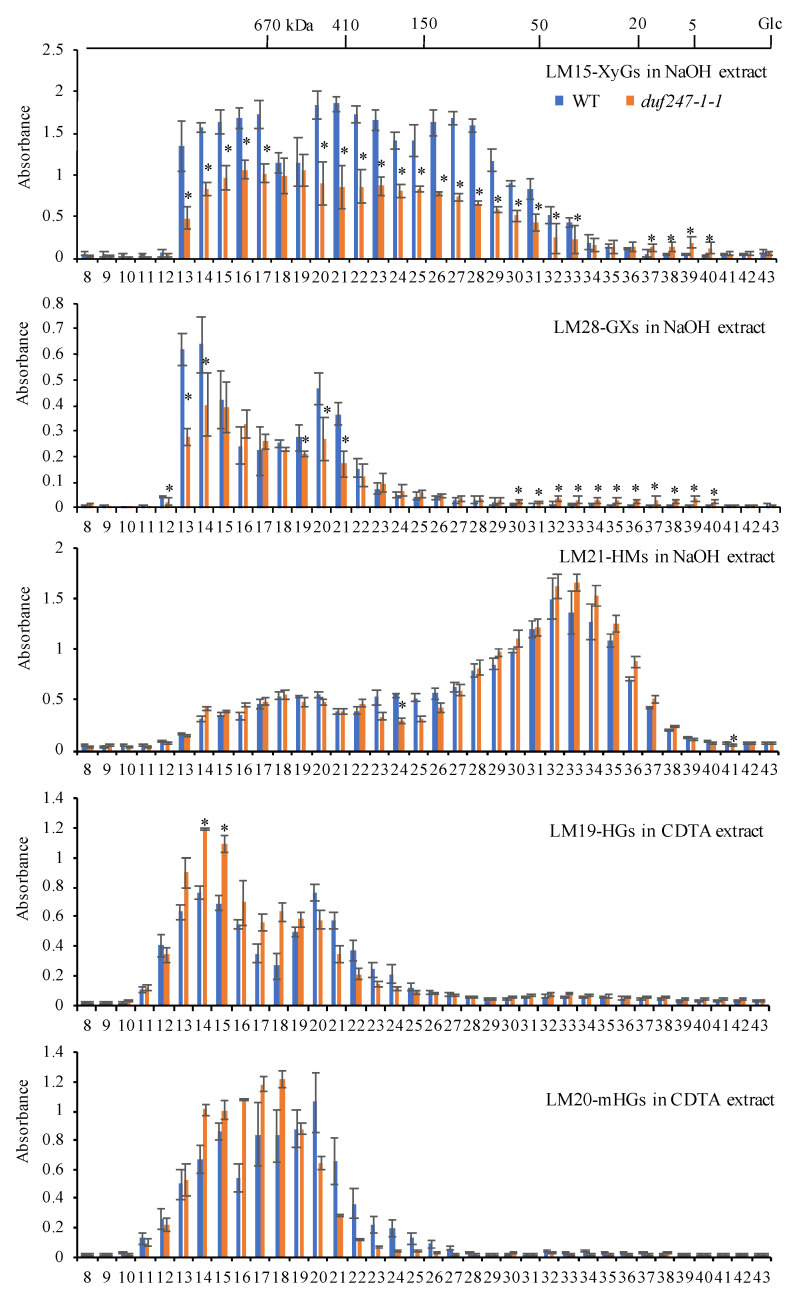
Mass distribution profiles of XyGs and GXs in *duf247-1-1* by GPC-ELISA. NaOH and CDTA extracts (2 mg mL^−1^) from the WT and *duf247-1-1* were fractionated using a Superose 6 Increase column. Fractions collected from NaOH extracts were used for ELISA using LM15, LM28 and LM21 mAbs for XyGs, GXs and HMs, respectively, and those from CDTA extracts were detected by LM19 and LM20 for HGs and mHGs, respectively. Data are presented as mean ± SE for three biological replicates and two technical replicates. The top *x*-axis represents mass markers (kDa) based on dextrans and glucose. Asterisks indicate significant differences at *p* < 0.05.

**Figure 6 plants-12-01977-f006:**
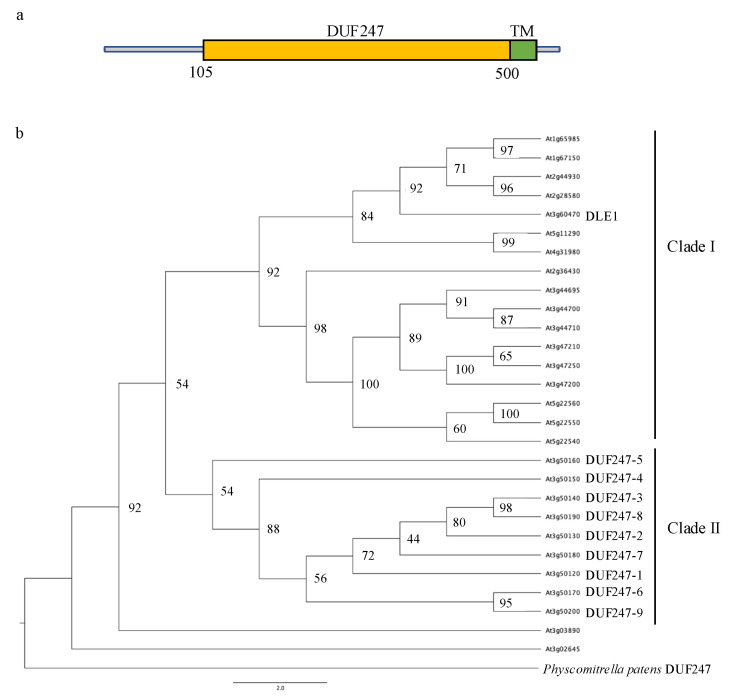
Structure and phylogenetic analysis of the DUF247 family. (**a**) Domain composition of DUF247-1 protein. Numbers indicate amino acid position. TM: predicted transmembrane protein. (**b**) The phylogenetic tree presenting an evolutionary relationship among the *Arabidopsis* DUF247 family. This phylogeny was built using the maximum likelihood method with the JTT + G + I + F model and 1000 bootstrap replicates.

**Figure 7 plants-12-01977-f007:**
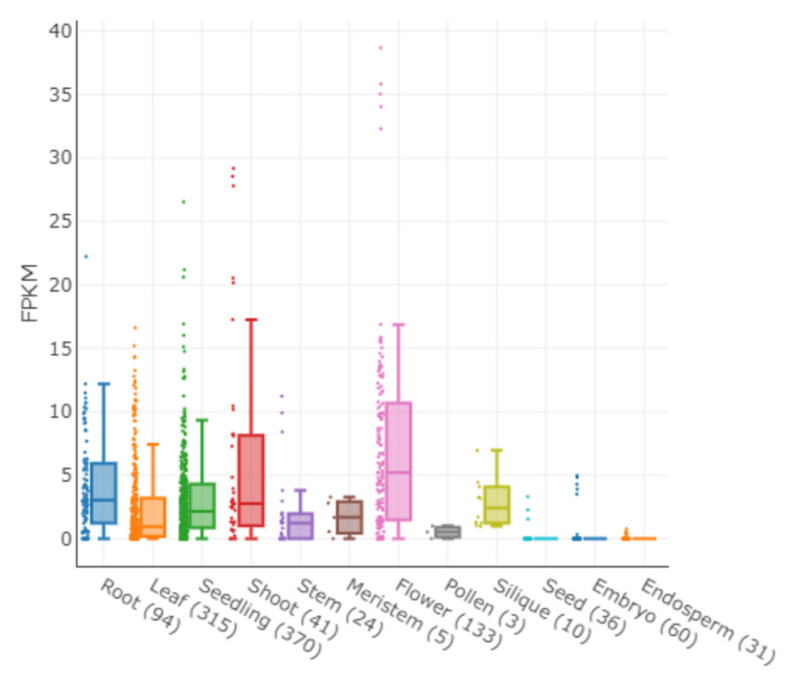
The expression profiles of *DUF247-1* in Arabidopsis tissues, obtained from Arabidopsis RNA-seq database (http://ipf.sustech.edu.cn/pub/athrdb/ accessed on 29 April 2023) for different tissues. The number of RNA-seq libraries used for each tissue is indicated in brackets.

**Figure 8 plants-12-01977-f008:**
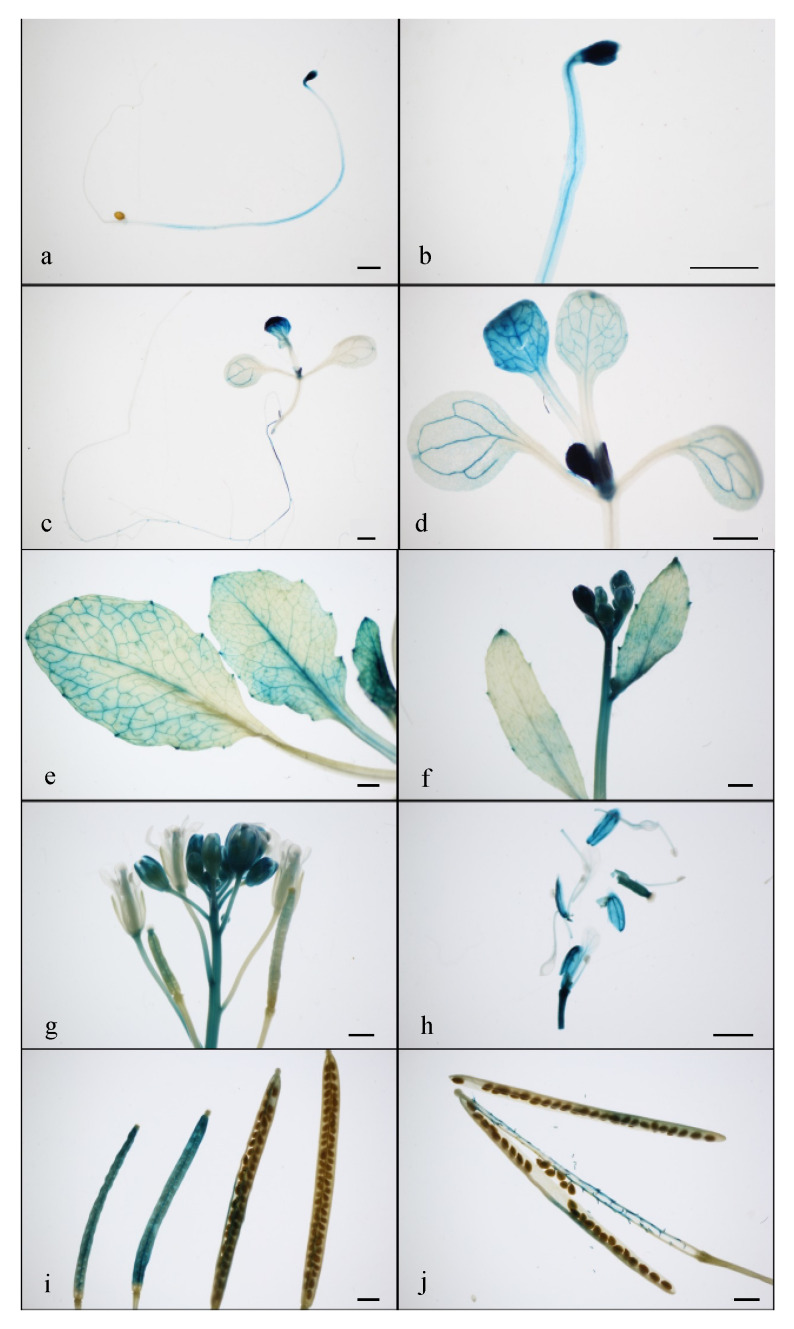
*DUF247-1* promoter–*GUS* fusion expression in *Arabidopsis*. Representative GUS activity observed in hypocotyl at 4 DAG (**a**,**b**), seedling at 7 DAG (**c**,**d**), rosette leaf (**e**), inflorescence (**f**,**g**), flowers (**h**), young and mature siliques (**i**,**j**). Scale bar = 1 mm.

**Figure 9 plants-12-01977-f009:**
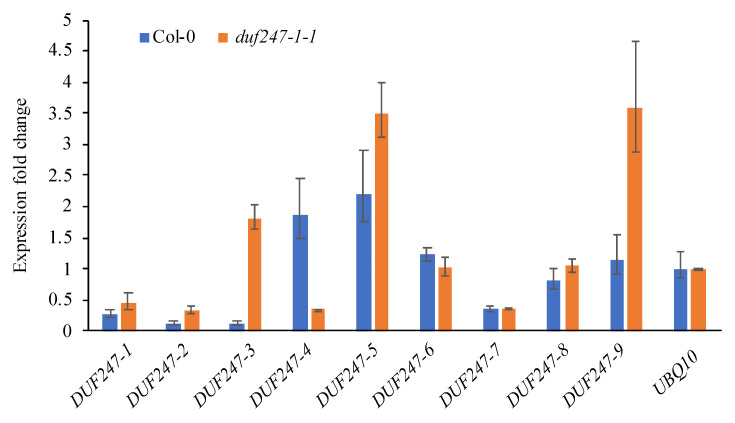
qRT-PCR analysis of nine *AtDUF247* genes of the DUF247 family clade II in 7-day-old seedlings of the WT and *duf247-1-1* mutant. Transcript levels are expressed in fold changes relative to that of *Ubiquitin10* (*UBQ10*). Data are presented as mean ± SE obtained from three biological replicates.

**Figure 10 plants-12-01977-f010:**
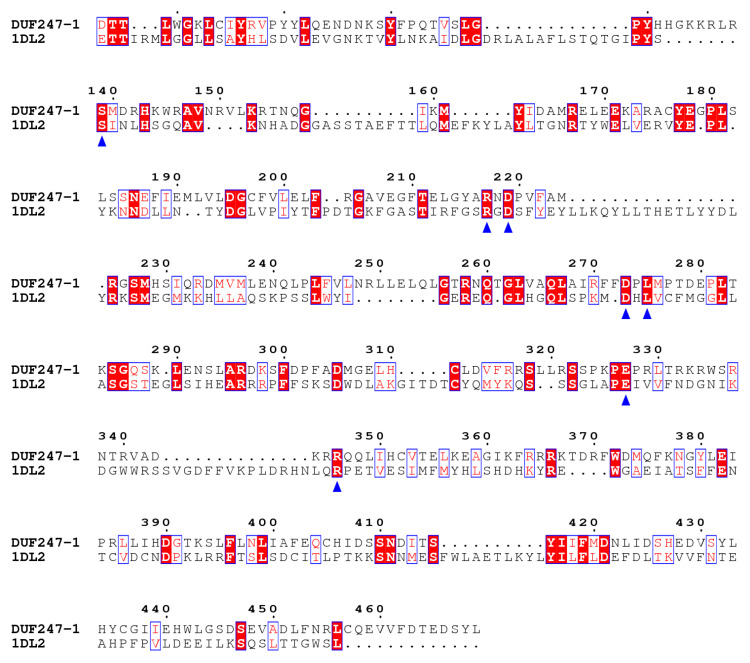
Amino acid sequence alignment of DUF247-1 without N- and C-terminal helices compared with yeast α-1,2-mannosidase (PDB ID: 1DL2). The residues binding with 1,2-, 1,3- and 1,6-mannose branches of *N*-glycan found in DUF247-1 are marked with triangles.

## Data Availability

All data generated and used in this study are available upon request or as Appendix A for this article.

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
