# Peer review of "Disruption of a DUF247 Containing Protein Alters Cell Wall Polysaccharides and Reduces Growth in *Arabidopsis"

_plants, 2023, doi:10.3390/plants12101977_

Round 1

Reviewer 1 Report

The manuscript of Wannitikul et al studied the potential function of a DUF247 protein in cell wall polysaccharide biosynthesis. This study is interesting and comprehensive. The authors performed good experiment design and properly organized the data. However, it will be better to improve two important results in Fig. 8 and Fig. 10. For the expression profile of DUF247-1, it is deserved to include the RT-qPCR result to compare the expression levels of DUF247-1 in different tissues. Observation of Golgi subcellular localization with Arabidopsis root cell is some rough which can’t recognize the Golgi pattern. The author can try tobacco leaf transient expression method. It will show the Golgi signal clearer.

1.       Fig. 1 legends are obviously disordered.

2.       Please describe the growth phenotypes of mature plants, too.

3.       Fig. 2b and 2c, the lines and words above the lines are misaligned with the below words.

4.       Fig. 2c, the primers used for RT pcr were not found in Fig. 2a.

5.       Line 292 to 295. It is better to include the predicted structure to supplemental figures.

6.       Fig. 11, the paragraph spacing is too large, please shrink them. The red color background is mismatched with letters at lots of places.

English writing is good.

Author Response

Dear Reviewers

Thank you very much for your time and effort reviewing our manuscript. We have addressed the points you raised and our answers are presented here.

Reviewer 1

The manuscript of Wannitikul et al studied the potential function of a DUF247 protein in cell wall polysaccharide biosynthesis. This study is interesting and comprehensive. The authors performed good experiment design and properly organized the data. However, it will be better to improve two important results in Fig. 8 and Fig. 10. For the expression profile of DUF247-1, it is deserved to include the RT-qPCR result to compare the expression levels of DUF247-1 in different tissues.

- We are grateful to the reviewer for the positive comments. In response to the comment about qRT-PCR, we have included expression data for DUF247-1 in various Arabidopsis tissues based on RNA-seq available in the Arabidopsis RNA-seq database (Zhang et al., 2020). The plant science community working with Arabidopsis has access to an large amount of data depicting gene expression in different organs and at many developmental stages. We have included the data available for our DUF247-1 gene and the results are presented as a new Figure 8. The results are described in P10, L240-245.

Observation of Golgi subcellular localization with Arabidopsis root cell is some rough which can’t recognize the Golgi pattern. The author can try tobacco leaf transient expression method. It will show the Golgi signal clearer.

- Regarding the localization experiment, we attempted to replicate it with Nicotiana benthamiana leaves, but unfortunately, clear signals for Golgi localization were not obtained. This is possible due to issues with microscopy or the experiment set-up that we used. As a result, we have decided to remove the localization section from the manuscript, as well as any claims about DUF247-1 being Golgi localized.

Specific points raised by the referee:

1. Fig. 1 legends are obviously disordered.

- We have modified the legend to follow the content in the Figure 1.

2. Please describe the growth phenotypes of mature plants, too.

- We have added a description of the growth phenotype of mature plants in P3, L95.

3. Fig. 2b and 2c, the lines and words above the lines are misaligned with the below words.

- We have aligned the lines with the words in the modified Figure 2.

4. Fig 2c, the primers used for RT pcr were not found in Fig. 2a.

- Primers used for the RT-PCR had been indicated in Materials and Methods section. We used the genotyping primers for RT-PCR. Here, we have indicated the primer names for RT-PCR in the legend of Figure 2.

5. Line 292 to 295. It is better to include the predicted structure to supplemental figures.

- The predicted structure of DUF247-1 has been included in Figure S2.

6. Fig. 11, the paragraph spacing is too large, please shrink them. The red color background is mismatched with letters at lots of places.

- We have reduced the spacing in Figure 11. The mismatched background was due to file formatting, and we have reformatted the Figure for correct labelling.

Reviewer 2 Report

The authors provide data to illustrate the function of a protein containing DUF47 domain on pectins in cell wall and growth of Arabidopsis. Generally, the results are new and well interpreted, however, there are still some points requiring further clarification. Below are some specific concerns:

Figure 1, symbols for significance of differences should be marked on lines connecting two groups for comparison, better with P value.

Do you have any explanation for the difference in hypocotyl length between the two knockout lines?

Abbreviations should be provided with the full terms at first appearance in the manuscript, e.g. CDTA.

FTIR spectra should be exhibited by differential spectra by substracting the results of mutant group from the WT group, which may display the differences clearly.

Again, symbols for significance of differences should be marked on lines connecting two groups in Fig. 3 and Fig. 5.

As Golgi bodies show punctate structures, the localization of Got1-GFP in Fig. 10 seems not a typical Golgi localized signal. Please check on the marker line and provide a new image.

Language polishing is required to increase the clarity and accuracy of sentences.

JIM series antibodies are also alternative choices to examine pectin related components. It would be better to have these view-directing results. I can understand the situation for adding more experiments, at least the results using these antibodies should be discussed.

Language polishing is required to increase the clarity and accuracy of sentences.

Author Response

Dear Reviewers

Thank you very much for your time and effort reviewing our manuscript. We have addressed the points you raised and our answers are presented here.

Reviewer 2

Comments and Suggestions for Authors

The authors provide data to illustrate the function of a protein containing DUF47 domain on pectins in cell wall and growth of Arabidopsis. Generally, the results are new and well interpreted, however, there are still some points requiring further clarification. Below are some specific concerns:

Figure 1, symbols for significance of differences should be marked on lines connecting two groups for comparison, better with P value.

- We have added lines connecting compared groups and P values.

Do you have any explanation for the difference in hypocotyl length between the two knockout lines?

- We performed the analysis for hypocotyl lengths of WT, two mutant lines and complemented lines at the same time. We don’t have a direct explanation for the difference between the two knockout lines. It is unlikely that it would be a result of different localization of the insertion as both lines are knockouts. The media used was the same, and the growing conditions were exactly the same. We have added this point to the text at P3, L104 as follows; “Noting that the hypocotyl lengths were different between the mutant lines, and it is unclear if this is a result of different T-DNA insertion positions.”

Abbreviations should be provided with the full terms at first appearance in the manuscript, e.g. CDTA.

- We have added the full terms of following abbreviations: CDTA, TFA, FTIR, PACE and ELISA.

FTIR spectra should be exhibited by differential spectra by substracting the results of mutant group from the WT group, which may display the differences clearly.

- We have modified the Figure by presenting FTIR spectra using subtracted profiles.  

Again, symbols for significance of differences should be marked on lines connecting two groups in Fig. 3 and Fig. 5.

- We have added lines connecting compared groups and P values in these Figures.

As Golgi bodies show punctate structures, the localization of Got1-GFP in Fig. 10 seems not a typical Golgi localized signal. Please check on the marker line and provide a new image.

- As explained above to Referee 1, we have decided to remove the localization section from the manuscript, as well as any claims about DUF247-1's Golgi localization.

Language polishing is required to increase the clarity and accuracy of sentences.

- The manuscript has undergone language proofreading to enhance sentence clarity and accuracy.

JIM series antibodies are also alternative choices to examine pectin related components. It would be better to have these view-directing results. I can understand the situation for adding more experiments, at least the results using these antibodies should be discussed.

- As suggested, we have added a discussion about alternative choices of antibodies in P14, L360-365.

Reviewer 3 Report

I enjoyed reading this and would like to commend the authors on a clearly written manuscript describing an interesting body of work.

My main concern is about the use of repeated t-tests to determine statistical significance. I cannot claim to be a statistical expert, but my understanding is that use of multiple pairwise comparisons like this increases the likelihood of false positives (type I errors). Again, I cannot claim to be an authority but I would usually use an ANOVA across all of the data to determine whether there are any significant differences, and then individual Tukey’s HSD tests to determine which differences are and at what level (although I would suggest checking with a statistician rather than following my lead).

I think that this is unlikely to affect the conclusions from Fig 1 e) and f) because there are few tests and the levels of significance are very high, but the Fig. 3 data look more marginal. In fact, because of this and some of the odd anomalies (for example why are the Complement GalA levels in the hot water/CDTA fraction higher than those in the mutants, which are higher than those in the wild type. On balance, I would suggest dropping the data from Fig. 3 as it looks inconclusive and does not contribute much to the overall conclusions. If it is retained, why is there no data for the complement in a) and b)?

Other more minor points:

1.       There is a sort of hybrid Harvard/numerical citation in line 59. I wouldn’t object but the editor may wish to advise.

2.        The differences between the mutants and wild type and complement plants in Fig. 1c are not terribly clear. On another matter, do the authors have any ideas about why duf247-1-2 had a lesser effect on hypocotyl growth than duf247-1-1?

3.       The FTIR data in Fig. 4 (which suggest lower levels of pectin) and the ELISA data in Fig. 5 d) and e) and Fig. 6  d) and e) (which suggest higher levels) seem slightly at odds. Do the authors have any views about why this could be?

4.       Might it be worth citing sources on the xxt1/xxt2 mutants (e.g. https://doi.org/10.1111/tpj.15666 ) as these show some similarities (reduced XG and increased pectins)?

Author Response

Thank you very much for your time and effort reviewing our manuscript. We have addressed the points you raised and our answers are presented here.

My main concern is about the use of repeated t-tests to determine statistical significance. I cannot claim to be a statistical expert, but my understanding is that use of multiple pairwise comparisons like this increases the likelihood of false positives (type I errors). Again, I cannot claim to be an authority but I would usually use an ANOVA across all of the data to determine whether there are any significant differences, and then individual Tukey’s HSD tests to determine which differences are and at what level (although I would suggest checking with a statistician rather than following my lead).

- As suggested, we have analyzed the data using ANOVA and Tukey’s HSD test and presented differences in groups. Figure 1 and 4 and their legends have been modified accordingly.

I think that this is unlikely to affect the conclusions from Fig 1 e) and f) because there are few tests and the levels of significance are very high, but the Fig. 3 data look more marginal. In fact, because of this and some of the odd anomalies (for example why are the Complement GalA levels in the hot water/CDTA fraction higher than those in the mutants, which are higher than those in the wild type. On balance, I would suggest dropping the data from Fig. 3 as it looks inconclusive and does not contribute much to the overall conclusions. If it is retained, why is there no data for the complement in a) and b)?

- We have removed Figure 3a and 3b and theirs content in the result 2.2. The monosaccharide composition analysis in Figure 3c (with ANOVA and Turkey HSD test) has been moved to Supplementary Figure S1. We think it is important to keep the monosaccharide data for readers to evaluate the effects of DUF247-1 disruption on cell wall polymers.   

Other more minor points:

1. There is a sort of hybrid Harvard/numerical citation in line 59. I wouldn’t object but the editor may wish to advise.

- This sentence has been rephrased to follow the numerical citation format.

2. The differences between the mutants and wild type and complement plants in Fig. 1c are not terribly clear. On another matter, do the authors have any ideas about why duf247-1-2 had a lesser effect on hypocotyl growth than duf247-1-1?

- The differences between WT, mutants and complement plant are more obvious during the seedling stage. The rosettes of the two mutants are clearly smaller than WT and complement lines during the first two weeks, while those of the mature stage were slightly smaller then WT and complement lines. This has been described in the Result section 2.1.  

- For hypocotyl length, we performed the analysis for WT, two mutant lines and complemented lines at the same time. We don’t have a direct explanation for the difference between the two knockout lines. It is unlikely that it would be a result of different localization of the insertion as both lines are knockouts. The media used was the same, and the growing conditions were exactly the same. We have already noted in the text that we aware of this difference between the two mutant as follows; “Noting that the hypocotyl lengths were different between the mutant lines, and it is unclear if this is a result of different T-DNA insertion positions.”

3. The FTIR data in Fig. 4 (which suggest lower levels of pectin) and the ELISA data in Fig. 5 d) and e) and Fig. 6  d) and e) (which suggest higher levels) seem slightly at odds. Do the authors have any views about why this could be?

- We have modified the text in Discussion section to address this point at P13L407 as follows: “Although the increases in HG epitopes shown by ELISA and GPC-ELISA are consistent with the increases in GalA in the hot water-CDTA fraction from both mutants, they are in contrast to the FTIR data that showed general decreased in pectins. Furthermore, the reductions of XyGs, GXs and HMs are not reflected in the monosaccharide composition in the TFA fraction. To the best of our knowledge, we are unable to implicate the monosaccharide composition and FTIR data further. The structures of matrix polysaccharides are complex structure, and ELISA results may not fully address the changes in these matrix polysaccharides due to limitation of the antibodies based on specific glycan structures.”

4. Might it be worth citing sources on the xxt1/xxt2 mutants (e.g. https://doi.org/10.1111/tpj.15666 ) as these show some similarities (reduced XG and increased pectins)?

- Based on Cavalier et al., (2008), xxt1 xxt2 double mutant had none detectable xyloglucan, but had no effect on pectins. So, we did not cover this suggestion in the text. Nonetheless we found that there are several aspect of the xxt1xxt2 double mutant that is similar to our mutant for reduction of xyloglucan and reduced growth and plant size. We add the following text in Discussion section.

“Given that the complementation restored normal growth and XyG and HG content, it is potentially the case that the reduction in XyGs is the cause of the growth phenotype. The reduced plant size and growth were also observed in the xxt1 xxt2 mutant that has no detectable XyG [47]. However, it is unclear whether the growth phenotype directly resulted from the altered XyG content or whether it was a collective contribution from changes in hemicelluloses and HGs.”

Round 2

Reviewer 1 Report

The authors have addressed all my comments in a satisfactory manner.

Author Response

We'd like to thank for your time evaluate our manuscript. 

Reviewer 2 Report

The authors have addressed most of my previous concerns, however, I found that the differential wavenumbers as revealed by differential spectra of FTIR were not discussed. The biological significance for these differential peaks should be discussed in the specific cellular comtext for this DUF247 protein. I have no further comment but this suggestion.

Author Response

Thank you very much for your time and effort reviewing our manuscript. We have addressed the point you raised here.

-  As suggested, we have added the context of FTIR result to Discussion section at P13L377-380 as follows:

“The wall alterations shown in this work are mostly consistent between the two independent duf247-1 mutants and are supported by complementation experiments. A broad reduction in FTIR profiles between 800-1,800 wavenumbers corresponding to cellulose, hemicellulose and pectins implicates a general decrease in cell wall polysaccharides, and this reflects that DUF247-1 could play a role in plant cell wall biosynthesis.”